# SARS-CoV-2 Neutralization Capacity in Hemodialysis Patients with and without a Fifth Vaccination with the Updated Comirnaty Original/Omicron BA.4-5 Vaccine

**DOI:** 10.3390/vaccines12030308

**Published:** 2024-03-15

**Authors:** Bo-Hung Liao, Louise Platen, Myriam Grommes, Cho-Chin Cheng, Christopher Holzmann-Littig, Catharina Christa, Bernhard Haller, Verena Kappler, Romina Bester, Maia Lucia Werz, Eva Platen, Peter Eggerer, Laëtitia Tréguer, Claudius Küchle, Christoph Schmaderer, Uwe Heemann, Lutz Renders, Ulrike Protzer, Matthias Christoph Braunisch

**Affiliations:** 1Institute of Virology, TUM School of Medicine and Health, Technical University of Munich, 81675 Munich, Germany; 2Department of Nephrology, TUM School of Medicine and Health, University Hospital Rechts der Isar, Technical University of Munich, 81675 Munich, Germany; 3TUM Medical Education Center, TUM School of Medicine and Health, Technical University of Munich, 81675 Munich, Germany; 4Institute of AI and Informatics in Medicine, TUM School of Medicine and Health, Technical University of Munich, 81675 Munich, Germany; 5Kidney Center Eifel Dialyse, 53894 Mechernich, Germany; 6KfH Kidney Center Harlaching, Munich-Harlaching, 81545 Munich, Germany; 7KfH Kidney Center, 83278 Traunstein, Germany; 8German Center for Infection Research (DZIF), Partner Site, 81675 Munich, Germany; 9Institute of Virology, Helmholtz Munich, 85764 Munich, Germany

**Keywords:** hemodialysis, SARS-CoV-2, updated Comirnaty Original/Omicron BA.4-5 vaccine, in vitro viral neutralization, Omicron BA.5, BQ.1.1, XBB.1.5

## Abstract

Background: Hemodialysis patients have reduced serologic immunity after SARS-CoV-2 vaccination compared to the general population and an increased risk of morbidity and mortality when exposed to SARS-CoV-2. Methods: Sixty-six hemodialysis patients immunized four times with the original SARS-CoV-2 vaccines (BNT162b2, mRNA-1273) either received a booster with the adapted Comirnaty Original/Omicron BA.4-5 vaccine 8.3 months after the fourth vaccination and/or experienced a breakthrough infection. Two months before and four weeks after the fifth vaccination, the live-virus neutralization capacities of Omicron variants BA.5, BQ.1.1, and XBB.1.5 were determined, as well as neutralizing and quantitative anti-SARS-CoV-2 spike-specific IgG antibodies. Results: Four weeks after the fifth vaccination with the adapted vaccine, significantly increased neutralizing antibodies and the neutralization of Omicron variants BA.5, BQ.1.1, and XBB.1.5 were observed. The increase was significantly higher than after the fourth vaccination for variants BQ.1.1 and BA.5. Of all analyzed variants, BA.5 was neutralized best after the fifth vaccination. We did not see a difference in humoral immunity between the group with an infection and the group with a vaccination as a fifth spike exposure. Fivefold-vaccinated patients with a breakthrough infection showed a significantly higher neutralization capacity of XBB.1.5. Conclusion: A fifth SARS-CoV-2 vaccination with the adapted vaccine improves both wild-type specific antibody titers and the neutralizing capacity of the current Omicron variants BA.5, BQ.1.1, and XBB.1.5 in hemodialysis patients. Additional booster vaccinations with adapted vaccines will likely improve immunity towards current and original SARS-CoV-2 variants and are, therefore, recommended in hemodialysis patients. Further longitudinal studies must show the extent to which this booster vaccination avoids a breakthrough infection.

## 1. Introduction

Patients on hemodialysis have an impaired innate and adaptive immune system due to uremia and frequent contact with dialysis membranes, resulting in reduced vaccination success and ability to fight infections [1]. This is equally valid for infections and vaccinations against SARS-CoV-2 [2,3]. The necessity to receive dialysis treatment several times a week hampers patients’ ability to maintain social distance and puts them at an increased risk for infection [4]. During the COVID-19 pandemic, hemodialysis patients were prioritized for vaccination when vaccines became available in Germany in 2021 [5].

A recently published study by Wijkström et al. demonstrated the increased morbidity and all-cause mortality of patients on renal replacement therapy in a large Swedish cohort during the first year of the pandemic when no vaccinations were available [6]. While all-cause mortality in kidney transplant patients remained increased after vaccines were available, in patients on renal replacement therapy, it was comparable to pre-pandemic times [6]. This underscores the importance of consistent immunization of hemodialysis patients and the close monitoring of the immunization success, especially since humoral immunity against COVID-19 is declining over time even after repeated vaccination [7].

Due to the novelty of the virus and the rapidly changing availability of vaccines and data on vaccination success, recommendations of vaccination schedules proposed by German authorities were constantly updated, and are currently recommending a basic immunization consisting of at least three SARS-CoV2-antigen exposures, two of which should be a vaccination with an approved vaccine as well as further booster vaccinations for indicated groups such as hemodialysis patients. Newly emerging SARS-CoV-2 variants with increased transmissibility or immune evasive properties pose a threat, especially to immunocompromised individuals such as hemodialysis patients. In the last year, Omicron BA.5, BQ.1.1, and XBB.1 subvariants have been prevalent in Germany, resulting in the recommendation for approval of adapted COVID-19 vaccines targeting, e.g., Omicron BA.4–5 and Omicron XBB.1.5 by the European Medicines Agency (EMA) [8,9,10,11]. A recent statement by the Standing Committee on Vaccination at the Robert Koch Institute (STIKO) in Germany reiterated their recommendation to perform a booster vaccination in hemodialysis patients, preferably with an adapted approved vaccine 12 months after the last antigen exposure, favorably administered in autumn [12]. Measuring patients’ live-virus neutralization of currently circulating variants is a laboratory method for estimating patients’ immune protection against these emerging variants. In this publication, we share results on hemodialysis patients’ humoral immunity against current SARS-CoV-2 variants Omicron BA.5, BQ.1.1, and XBB.1.5 after a fifth vaccination with the adapted Comirnaty Original/Omicron BA.4-5 vaccine.

## 2. Material and Methods

### 2.1. Study Design

The study design of the observational COVIIMP cohort (in German: “COVID-19-Impfansprechen immunsupprimierter Patient*innen”) has been described previously in detail [13]. In summary, immunocompromised patients are observed longitudinally regarding their serological and clinical response to SARS-CoV-2 immunization. The study is registered at Paul Ehrlich Institute (NIS592), and local ethics committee approval has been given as ethic vote 163/21 S-SR, on 19 March 2021, by the Medical Ethics Committee of the Klinikum rechts der Isar (Technical University of Munich). Patients were enrolled after providing written consent. The administration of vaccinations was not part of the study; patients were vaccinated by their treating physicians according to current local recommendations at that time.

### 2.2. Participants

A total of 66 hemodialysis patients were either fourfold vaccinated with a history of infection or received a fifth vaccination with the updated Comirnaty Original/Omicron BA.4-5 vaccine. Patients were followed up for 9.2 months after the fourth vaccination. Initially, 142 fourfold-vaccinated patients from four dialysis centers (Kidney Center Eifeldialyse Mechernich, Germany, KfH Kidney Center München-Harlaching, Munich, Germany, KfH Kidney Center Traunstein, Traunstein, Germany and Klinikum rechts der Isar München, Munich, Germany) were enrolled into the study. Inclusion in the study was offered to all present patients in two days in order to reach all patients of the dialysis center (Monday, Wednesday, Friday and Tuesday, Thursday, Saturday groups). Four patients were lost to follow-up due to kidney transplantation, 2 patients changed dialysis centers, 1 patient experienced a recovery of kidney function, 16 patients died, and 49 patients could not be followed up for logistic reasons. One patient was excluded for receiving a fifth vaccination with the original BNT162b2 vaccine, and 3 patients were neither vaccinated a fifth time nor infected. Finally, 66 hemodialysis patients remained for the current analysis.

The 66 hemodialysis patients were stratified into three groups according to their vaccination and infection status. In the first group (group A) n = 14 patients were vaccinated four times and had a history of SARS-CoV-2 infection. The second group (group B) of n = 32 patients had a fifth vaccination and naivety towards the SARS-CoV-2 virus. The third group (group C) of n = 20 patients had a fifth vaccination as well as a history of SARS-CoV-2 infection (Figure 1).

Clinical data including patient history, dialysis vintage, prescribed medication, vaccination status, and SARS-CoV-2 infections were collected, and blood sampling was performed four times in the period between March 2022 and December 2022 (Figure 2). The first blood analysis in the fivefold-vaccinated cohort (Group B and C, n = 52) was performed in median 2 (IQR 2–2) days before the 4th, corresponding to 124 (124–124) days after the 3rd vaccination (blood analysis 1 in Figure 2). Blood analyses after the 4th vaccination in this cohort were done in median 26 (26–26) days (blood analysis 2) and 191 (191–191) days (blood analysis 3) after the vaccination. For the 5th vaccination in the fivefold-vaccinated cohort, blood was drawn in median 28 (28–28) days afterwards (blood analysis 4). The first blood analysis in the fourfold-vaccinated cohort (Group A, n = 14) was performed in median 2 (2–2) days before the 4th vaccination, which corresponds to a median of 124 (108–125) days after the 3rd vaccination (blood analysis 1). Blood analyses after the 4th vaccination in Group A were done in median 26 (26–26) days (blood analysis 2) and 191 (191–191) days (blood analysis 3) after the vaccination. The last blood analysis in this group, which did not receive a 5th vaccination, was done in median 280 (280–282) days after the 4th vaccination (blood analysis 4).

### 2.3. Infections

Breakthrough infections were categorized based on their timing into “early”, “late”, and “very late” infections. “Early” infections occurred between May 2021 and February 2022, coinciding with the Delta wave in Germany (including the end of the Alpha and the beginning of the Omicron BA.1 wave) [14]. “Late” infections occurred between March 2022 and July 2022 when Omicron BA.2 and BA.5 were predominant in Germany [15]. In the period of “very late” infections between September and December of 2022, predominantly Omicron BA.5 infections were detected in Germany [16]. We registered double infections in three patients, with the first infection taking place prior to the study period in all three cases. One individual in the fourfold-vaccinated/infected group (group A) experienced a “very early” infection prior to the first vaccination and a “late” infection during the study period. Two patients in the fourfold-vaccinated/infected group (group C) with “very early” infections prior to and shortly after the first vaccination were infected a second time with “late” and “very late” infections during the observational period. Patients were considered infected with SARS-CoV-2 if a PCR or point-of-care testing (POCT) had been positive or if new nucleocapsid-specific IgG antibodies were detected in the blood analyses.

### 2.4. Serological Methods

Patients’ sera were obtained in serum separation tubes and centrifuged after collection. The serum was either measured within one day after collection or stored at −80 °C until analyzed. Antibody assays and virus neutralization capacities of VoC Omicron BA.5 and BQ.1.1 had been analyzed previously for some patients; missing analyses were added accordingly [7,13]. The measurement of anti-spike (S) IgG antibodies and neutralization capacities of VoC Omicron XBB.1.5 were performed in samples from blood analysis three and four.

### 2.5. Antibody Assays

Humoral immunity was assessed with three antibody assays, detecting SARS-CoV-2 nucleocapsid specific IgG type antibodies (anti-n IgG), SARS-CoV-2 receptor-binding domain specific neutralizing antibodies (NAb) and wild-type-spike-specific anti-SARS-CoV-2 IgG (anti-spike (S) IgG).

### 2.6. NAb and Anti-n IgG

The methods have been detailly described previously [7,13]. Antibodies were measured on the automated iFlash 1800 platform with a surrogate neutralization assay (CLIA, YHLO Biotechnology, Shenzhen, China). The assay for NAb measurement (Flash 2019-nCoV NAb kit) is based on the competition of patients’ antibodies and recombinant angiotensin-converting enzyme 2 for binding to a microparticle-coupled SARS-CoV-2 Wuhan strain receptor-binding domain. Titers are given in BAU/mL with an upper and lower limit of quantification of 800 and 4 BAU/mL. The 2019-nCoV IgG kit was used for the measurement of anti-n-IgG. NAb and anti-n-IgG were considered positive at values ≥ 10 (B)AU/mL.

### 2.7. Anti-Spike (S) IgG

The Abbott SARS-CoV-2 IgG II Quant chemiluminescent microparticle immunoassay for measuring anti-spike (S) IgG was performed on the Abbott Architect i1000SR platform (Abbott, IL, USA) according to the manufacturer’s instructions. Microparticles coated with wild-type SARS-CoV-2 spike protein bind patient antibodies against this antigen. The bound antibodies then react with acridinium-labeled anti-human IgG conjugates. The chemiluminescence signal is measured in relative light units and is directly proportional to the amount of IgG antibodies bound to the S protein, including the RBD of SARS-CoV-2. The analytical measurement range is 21 to 40,000 AU/mL, and the threshold for seropositivity is ≥50 AU/mL. AU/mL can be converted to Binding Antibody Units (BAU) per milliliter by using a conversion factor (BAU/mL = AU/mL × 1/7) [17]. Statistical measurements at the limits of quantification were done with a value of 4 and 801 for NAbs and 21 and 40,001 BAU/mL for anti-spike (S) IgG.

### 2.8. Live-Virus Neutralization Assays

The neutralization capacity of patients’ sera was analyzed as described previously [7,13]. Variants of concern, including Omicron BA.5 (GISAID EPI ISL: 15942298), BQ.1.1 (GISAID EPI ISL: 15812430), and XBB.1.5 (GISAID EPI ISL: 17300038), which had been acquired from nasopharyngeal swabs of infected individuals were incubated with Vero E6 cells in Dulbecco’s Modified Eagle’s Medium (DMEM). After two to three days, the supernatant was collected. The high titer viral stock was stored at −80 °C until the previously defined MOI (multiplicity of infection) of 0.03; plaque-forming units/cell (450 PFU/15,000 cells/well) were incubated at 37 °C for one hour with the dilutions of patients’ sera at 1:20 to 1:2560. Next-generation sequencing was performed to verify viral strains, and plaque assays were done to determine viral titers. The inoculum was incubated for one hour with Vero E6 cells seeded into 96-well plates at the same temperature until the inoculum was removed and the cells were washed. After fixing and permeabilizing the inoculated Vero E6 cells with 4% paraformaldehyde and 0.5% saponin buffer, cells were blocked with 10% goat serum and stained with a primary anti-SARS-CoV-2-N antibody (40143-T62, Sino Biological, Beijing, China). Goat anti-rabbit IgG2a-HRP secondary antibody (12-348, EMD Millipore, Shanghai, China) and substrate TMB (tetramethybezidine) were used for in-cell ELISA colorimetric analysis. IC50 values (50% inhibitory serum concentration) were defined as a dilution factor with 50% infection inhibition by nonlinear regression calculated with Graph Pad Prism Version 9.5.1 (528) (GraphPad Software, San Diego, CA, USA). An IC50 of ≤1:20 was considered as a non-response.

### 2.9. Statistics

Statistical analysis was performed with Graph Pad Prism Version 9.5.1 (GraphPad Software, San Diego, CA, USA) and R version 4.2.1 (R Foundation for Statistical Computing, Vienna, Austria). Categorial variables are presented as numbers and percentages; continuous variables are given as the median and interquartile range (IQR). Groupwise comparison was tested with the Wilcoxon test for connected samples and the Mann–Whitney test for unpaired samples, or the χ^2^ test as appropriate. For multiple tests, the Friedman test was performed for connected samples and the Kruskal–Wallis Test for unpaired samples. Dunn’s multiple comparisons test was used as a post hoc test corrected for multiple comparisons. The Spearman test was done for correlation analysis. The significance level was set at *p* < 0.05.

For calculating the vaccination effect (“delta”) of the 4th and 5th vaccination, the individual IC50 value after vaccination was subtracted from the respective value before the vaccination.

## 3. Results

### 3.1. Patient Characteristics

Overall, 66 patients on hemodialysis were enrolled in this study and followed up for a median of nine months (282 (282–282) days) after the first blood analysis (before the fourth COVID-19 vaccination) (Figure 1 and Figure 2). The median age was 72.7 (60.3–81.5) years, and 21 of the 66 (31.8%) patients were female. Dialysis vintage at baseline was 42.0 (15.5–65.5) months. Immunosuppressive medication was present in 10 individuals. One patient in the fourfold-vaccinated/infected group (group A) received tacrolimus, mycophenolate mofetil, and prednisolone. Another patient received tacrolimus and prednisolone in both cases due to a history of kidney transplantation. In the fivefold-vaccinated/naïve group (group B), five patients received immunosuppressive medication, indicated by a history of kidney transplantation in two cases (tacrolimus with prednisolone and prednisolone only). Three patients received prednisolone due to anterior ischemic optic neuropathy, vasculitis, or lupus nephritis. In the fivefold-vaccinated/infected group (group C), three patients received immunosuppressive medication due to a history of lung transplantation in one patient (tacrolimus, mycophenolate mofetil, prednisolone), due to multiple myeloma in one (lenalidomide and prednisolone), and a third patient received prednisolone due to an unknown cause. A higher dialysis vintage was observed in the fivefold-vaccinated/naïve group (group B) compared to the fourfold-vaccinated/infected group (group A) (Table 1).

### 3.2. Vaccinations

All 66 patients received mRNA-based SARS-CoV-2 vaccines only. All 52 fivefold-vaccinated patients received the updated Comirnaty Original/Omicron BA.4-5 vaccine as the fifth vaccination. One patient received mRNA-1273 by Moderna as the first and second vaccination, and the rest were performed with BNT162b2 by BioNTech Pfizer.

### 3.3. SARS-CoV-2 Infections

Patients were divided into three groups (A, B, and C) according to their vaccination and infection status (Figure 1). The fourfold-vaccinated/infected group (group A) consists of 14 patients who had experienced a SARS-CoV-2 breakthrough infection. Twelve of these infections were PCR/POCT-confirmed. Two of these infections were solely detected by positive n-specific IgG status. One of these patients had been infected previously before the first blood analysis.

The fivefold-vaccinated/infected group (group C) consists of 20 individuals. Ten of these individuals’ breakthrough infections were solely seen by positive n-specific IgG status; the other ten were PCR/POCT-confirmed. Two patients had been infected before the study period as these infections occurred before the first blood analysis.

Three infections in the fourfold-vaccinated/infected group (group A) and five infections in the fivefold-vaccinated/infected group (group C) were classified as “early” infections. They occurred between May 2021 and February 2022, when the Delta VoC was predominant in Germany (plus the end of the alpha and the beginning of the Omicron BA.1 wave). Three infections in the fourfold-vaccinated group and eight infections in the fivefold-vaccinated group were defined as “late” infections. These occurred between March 2022 and July 2022 during Germany’s Omicron BA.2 and BA.5 wave. There were eight “very late” infections that occurred between September and December of 2022, when Omicron BA.5 was predominant in Germany in the fourfold-vaccinated/infected group (group A) and seven “very late” infections in the fivefold-vaccinated/infected group (group C) (Figure 2).

### 3.4. Effect of the Fourth and Fifth Vaccination

Immunity of hemodialysis patients in the fivefold-vaccinated cohort (group B and C, n = 52) before and after the fifth vaccination with the updated Comirnaty Original/Omicron BA.4-5 vaccine is displayed in Table 2 and Figure 3. There was a significant increase in neutralizing antibodies, anti-spike (S) IgG, and live-virus neutralization capacities of all tested Omicron variants BA.5, BQ.1.1, and XBB.1.5 after the fifth vaccination with the adapted Comirnaty Original/Omicron BA.4-5 vaccine. In the fivefold-vaccinated cohort (group B and C, n = 52), the neutralization capacity of BA.5 was significantly higher compared to the neutralization capacity of BQ.1. and XBB.1.5. This was true before and after the fifth vaccination. When comparing the delta of the increase, we observed a significantly higher increase for the XBB.1.5 compared to the BQ.1.1 neutralization capacity (Figure 4A,B). The effect of the fifth vaccination on the neutralization capacity of Omicron BA.5 and BQ.1.1 was compared to the impact of the fourth vaccination in infection-naïve hemodialysis patients (n = 32) (Figure 5). The timing of blood analysis after the vaccinations was comparable for the fourth and fifth vaccination. It was a median of 26 and 28 days apart, respectively. Blood analyses were conducted at a median of 124 days after the third vaccination and 191 days after the fourth vaccination. We detected a significantly greater increase in neutralization capacity for both VoCs after the fifth compared to after the fourth vaccination. To verify whether the increase was influenced by different baseline levels of neutralization capacities before the vaccinations, those baseline levels were compared. We did not see a significant difference for variant BQ.1.1 (*p* = 0.741); however, a trend of higher baseline values was seen before the fourth compared to before the fifth vaccination for variant BA.5 (*p* = 0.052). Median IC50 values before the fourth vaccination and before the fifth vaccination were 89.4 (20.0–588.7) and 67.35 (21.1–266.7) for BA.5 and 20.0 (0.0–20.0) and 20.0 (0.0–20.0) for BQ.1.1.

### 3.5. Effect of Vaccination and Breakthrough Infections

When performing a group-wise comparison of immunity between the three groups, we did not see a difference in antibody titers or neutralization capacities between fourfold-vaccinated plus infected and fivefold-vaccinated naïve patients (Table 2). After the fifth vaccination, we saw significantly higher anti-spike (S) IgG antibodies and neutralization capacity for XBB.1.5 in fivefold-vaccinated and infected patients compared to fourfold-vaccinated and infected patients. There were no differences detectable for BA.5 and BQ.1.1. When comparing fivefold-vaccinated/naïve (group B) and infected (group C) patients after the fifth vaccination, we saw a significantly higher virus neutralization of Omicron XBB.1.5 and BA.5 as well as, by trend, a higher neutralization capacity of BQ.1.1 and neutralizing antibodies. Significant differences were seen already before the fifth vaccination in some cases; some of the infections occurred before and some after this blood analysis, so results should not be over-interpreted (Table 2). With Kruskal–Wallis and a post hoc test for multiple comparisons, significant differences could only be observed for XBB.1.5, showing a significantly higher neutralization capacity in fivefold-vaccinated/infected patients (group C) compared to the two other groups (groups A and B) (Figure 6). Median results of antibody titers and neutralization capacities after the fifth vaccination are displayed in Table 3. Due to low numbers, statistical analysis of the subgroups stratified by the time of infections was difficult. When calculating the Spearman correlation, no significant correlation between the time of infection and neutralization capacities could be found.

## 4. Discussion

In this study, we show that hemodialysis patients experience a significant increase in the live-virus neutralization capacity of the recently circulating Omicron VoCs BA.5, BQ.1.1, and XBB.1.5 28 days after the fifth vaccination with the adapted bivalent Comirnaty Original/Omicron BA.4-5 vaccine. Also, SARS-CoV-2 Wuhan strain-specific neutralizing and quantitative antibodies significantly increased after the fifth vaccination, showing that a fifth vaccination with the adapted bivalent vaccine stimulates an immune response targeting the original Wuhan strain virus and current VoCs in hemodialysis patients. A recent study by Tani et al. that monitored hemodialysis patients’ COVID-19 vaccination response found large percentages of individuals with high Wuhan strain neutralizing and anti-spike (S) IgG antibodies already before the fifth vaccination to be increasing even further after the fifth vaccination with the adapted bivalent Comirnaty BA.4-5 vaccine [19]. Similar results were observed in a pseudovirus neutralization assay of the wild-type and Omicron BA.4-5 variants with a significantly increased cellular immunity after the bivalent vaccination (increased SARS-CoV-2 CD4+ and CD8+ T cell count and functionality against both wild-type and BA.4-5) [20]. Another recent publication by Huth et al. showed an increase in the neutralization of variants BA.4 and BA.5 in infection-naïve hemodialysis patients after the fifth vaccination with a bivalent BA.4-5-adapted vaccine [21]. This was also observed in a study by Benning et al., however, without an increase in humoral immunity in those patients with a prior SARS-CoV-2 infection [22].

Similar findings were seen in our cohort. Additionally, we analyzed the live-virus neutralization capacities of the more current variants BQ.1.1 and XBB.1.5, which were neutralized significantly less effectively than variant BA.5 before the fifth vaccination. When comparing the neutralization capacities of the different Omicron variants, BA.5 was neutralized significantly better than VoCs BQ.1.1 and XBB.1.5. This is consistent with previously published analyses for BA.5 and XBB.1.5 [23]. For Omicron BA.5, the most values at the upper limit of quantification were seen, even pointing to an underestimation of our calculations. Omicron BA.5 was already best neutralized at the blood analysis before the fifth vaccination, which may be explained by the fact that most infections in our cohort occurred during a time when Omicron BA.5 or earlier variants were predominant in Germany where BQ.1.1 had barely been detected and XBB.1.5 had not been detected yet. Nevertheless, we also saw a significantly higher increase in the neutralization capacity of BA.5 compared to both other VoCs with the fifth vaccination, which was expected since a BA.4-5-adapted vaccine was used.

We also observed a significantly higher increase in the VoC XBB.1.5 compared to BQ.1.1 after the fifth vaccination, which is of high relevance since the VoC XBB and its subvariants have been prevalent throughout the year 2023 and still are relevant VoCs in Germany, resulting in the recent recommendation for approval of adapted COVID-19 vaccines targeting Omicron XBB.1.5 by the European Medicines Agency (EMA) [8,9,10]. To assess the effect of the adapted in contrast to the original vaccine on recent variants, we compared the increase in infection-naïve patients’ neutralization capacities of VoCs BA.5 and BQ.1.1 after the fourth vaccination to the increase after the fifth vaccination, which was significantly greater after the fifth vaccination in both cases. The IC50 values before the vaccinations were comparable for BQ.1.1 and for BA.5; however, a trend of lower IC50 values before the fifth compared to before the fourth vaccination was seen. This and the greater distance to the previous antigen exposure before vaccination five must be considered as confounding factors. No significant difference in antibody titers or the neutralization of current variants was observed in the group-wise comparison of fourfold-vaccinated/infected patients (group A) to fivefold-vaccinated/naïve patients (group B). Fivefold-vaccinated/infected patients (group C), however, showed significantly better neutralization of variants BA.5 and XBB.1.5 compared to fivefold-vaccinated/naïve patients (group B), implying that breakthrough infections may still broaden immunity in comparison to naïve patients immunized with both the original and an adapted vaccine. Interestingly, the significant difference between these groups was only observed for variant XBB.1.5 in the multiple comparison analysis. This implies that patients may benefit from a further vaccination against these current variants, ideally with an adapted vaccine. Especially regarding XBB subvariants, like the now emerging EG.5.1 (XBB.1.9.2.5.1) with a higher effective reproduction number than XBB.1.5, which has been classified as a variant of interest in August, more research of patients’ serological and clinical immunity after vaccinations with the adapted vaccines is needed [24].

In the analysis of subgroups, no correlation could be found between patients’ immune response and the distance to the infection for patients with known infection dates. However, a weaning humoral immunity after infections has been shown before. As median values in Table 3 suggest, the distance between blood analysis and infection is likely to influence antibody titers and neutralization capacities [25,26].

Finally, some limitations must be discussed.

Due to the observational design of the study and the limited number of patients per group, there is a potential for bias and results may not be generalizable.

While XBB.1.5 infections have been predominant in Germany in May and June of 2023, the newly declared variant of interest EG.5 has been emerging in Germany. It has accounted for most infections since July 2023 [9]. While XBB.1.9.2, the variant that EG.5 descended from, has the same spike protein as XBB.1.5, EG.5 itself has a mutation in its spike protein, leading to immune-evasive properties and possible advantages in transmissibility [27]. Therefore, further studies on the effectiveness of vaccinations against this current variant of interest are needed.

Neutralization capacity is known to correlate with SARS-CoV-2 infection rates and COVID-19 disease severity [28]. However, besides the capacity of neutralizing antibodies, the vaccine-induced immunity, particularly the disease severity, also depends on T cell-mediated immune mechanisms [29,30]. We did not analyze cell-mediated immunity or infection rates and disease severity in this study. However, a solid cell-mediated immunity against BA.4-5 has been shown in hemodialysis patients after a bivalent fifth vaccination [20]. Whether this immunity extends to the more current variants remains to be seen.

We did not assess the dialytic membranes used; however, recent studies have focused on the role of different dialytic membranes in the altered immune status and vaccination success in hemodialysis patients [31]. Further research is needed to find factors associated with an improved vaccination response in hemodialysis patients.

## 5. Conclusions

The data suggest that a fifth vaccination with an adapted COVID-19 vaccine improves humoral immunity against both original and evolved SARS-CoV-2 strands in fourfold-vaccinated hemodialysis patients. Compared to the fourth vaccination, the fifth vaccination induces a more significant increase in the neutralization capacities of VoCs BA.5 and BQ.1.1. Out of the three tested variants, BA.5 is neutralized best both before and after the fifth vaccination and shows the most significant increase in neutralization after the fifth vaccination. No significant difference in antibody titers or the neutralization of current variants was observed when comparing fourfold-vaccinated infected patients to fivefold-vaccinated naïve patients. However, fivefold-vaccinated and infected patients showed significantly better neutralization of the newly emerged variants BA.5 and XBB.1.5 than fivefold-vaccinated and virus naïve patients. In conclusion, a fifth vaccination with an adapted vaccine seems sensible in hemodialysis patients who did not experience a spike exposition during the eight months following the fourth vaccination.

## 6. Key Learning Points

### 6.1. What Is Already Known about This Subject?

Hemodialysis patients are impaired in their immune response against SARS-CoV-2 and have an elevated morbidity and mortality risk compared with the general population.COVID-19 vaccinations activated immunity against SARS-CoV-2 in hemodialysis patients, but this is weaning over time, necessitating monitoring of immune status and regular booster vaccinations.Emerging new variants of SARS-CoV-2 usually possess immune-evasive properties and increased transmissibility, requiring the regular analysis of patients’ immunity against these new variants.

### 6.2. What Does This Study Add?

In hemodialysis patients, a fifth vaccination with a BA.4-5-adapted COVID-19 vaccine leads to an improved humoral immunity against the original and evolved SARS-CoV-2 strains eight months after the last vaccination.The increase in neutralizing capacity after the vaccination with the BA.4-5-adapted vaccine is significantly lower for more recent VoCs like BQ.1.1 and XBB.1.5 compared to BA.5.Humoral immunity towards the original and recent SARS-CoV-2 variants did not differ between fourfold-vaccinated HD patients with a recent history of infection and SARS-CoV-2-naïve patients after a fifth vaccination with an adapted COVID-19 vaccine.

### 6.3. What Impact May This Have on Practice?

A fifth vaccination seems sensible in hemodialysis patients who did not experience a spike exposition within eight months following the fourth vaccination.The exposition to a SARS-CoV-2 spike protein of more recent VoCs, e.g., in an XBB-adapted vaccine, might provide patients with better immunity against current variants.Comparable to COVID-19 vaccines, breakthrough infections provide patients with a broadened immunity against current VoCs.

## Figures and Tables

**Figure 1 vaccines-12-00308-f001:**
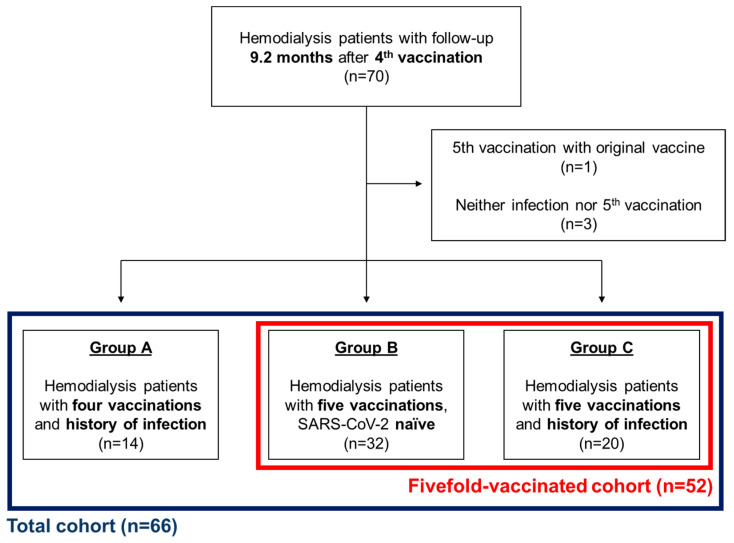
Flow chart. Sixty-six hemodialysis patients were included in the analysis of this work and were followed up for 9.2 months after the 4th COVID-19 vaccination. Fourteen individuals with a history of a SARS-CoV-2 breakthrough infection did not receive a 5th vaccination. Fifty-five patients received a 5th vaccination with the adapted Comirnaty Original/Omicron BA.4-5 vaccine (red). Naivety towards the SARS-CoV-2 virus was present in n = 32, whereas n = 20 individuals had experienced a breakthrough infection.

**Figure 2 vaccines-12-00308-f002:**
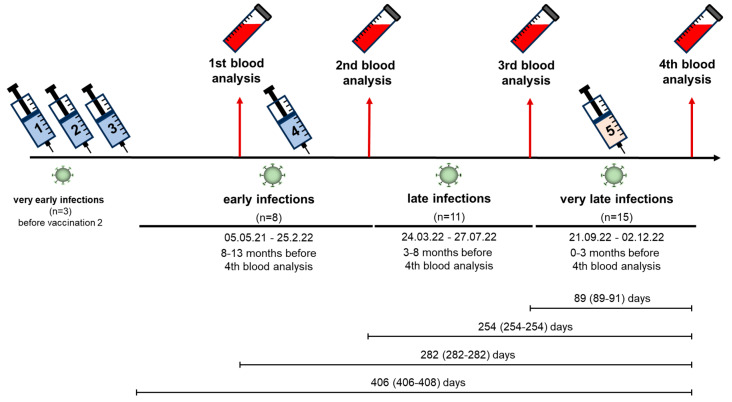
Study timeline with breakthrough infections in the whole cohort (n = 66). Breakthrough infections were stratified by their timing into “early”, “late”, and “very late”. “Early” infections occurred between May 2021 and February 2022, during Germany’s Delta wave (plus the end of the Alpha and the beginning of the Omicron BA.1 wave). “Late” infections occurred between March 2022 and July 2022 during the Omicron BA.2 and BA.5 wave in Germany. In the period of “very late” infections between September and December of 2022, Omicron BA.5 was predominant in Germany. Distances are given in days as Median (IQR).

**Figure 3 vaccines-12-00308-f003:**
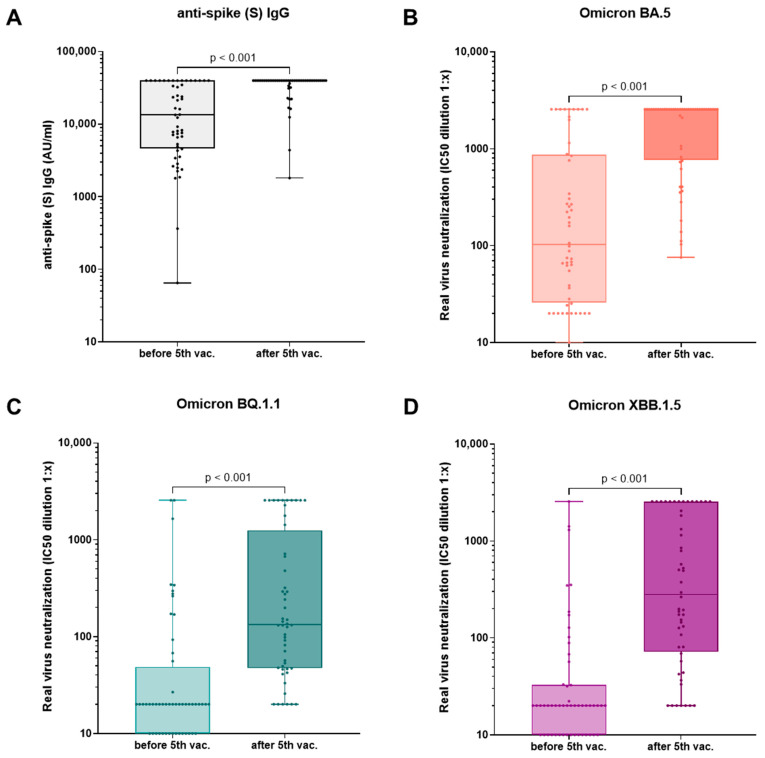
Humoral immunity before and after the 5th vaccination. Anti-spike (S) IgG (**A**) and live-virus neutralization capacities of the recently predominant Omicron variants in Germany BA.5 (**B**), BQ.1.1 (**C**), and XBB.1.5 (**D**) in hemodialysis patients who received a 5th vaccination with the updated Comirnaty Original/Omicron BA.4-5 vaccine (n = 52) before and after the 5th vaccination. After the 5th vaccination, neutralizing antibodies and live-virus neutralization of all analyzed variants were significantly increased. Antibody titers are expressed in AU/mL, and neutralization capacity is shown as dilution titers with 50% inhibition of infection (IC50). Statistical analysis was performed using Wilcoxon signed-rank test for paired samples. For improved clarity, a logarithmic scale was chosen for the y-axis. Abbreviations: vac., vaccination.

**Figure 4 vaccines-12-00308-f004:**
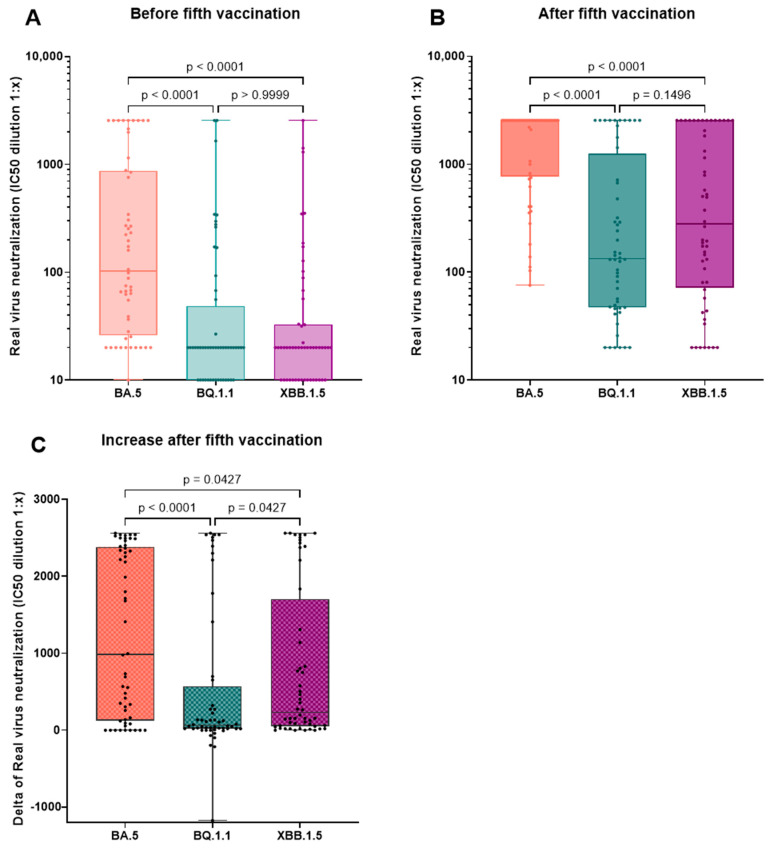
Comparison of virus neutralization capacities of Omicron variants BA.5, BQ.1.1, and XBB.1.5 before and after the 5th vaccination and the change after the 5th vaccination. Neutralization capacity of the different variants before (**A**) and after (**B**) the 5th vaccination with the bivalent adapted Comirnaty Original/Omicron BA.4-5 vaccine in the fivefold-vaccinated cohort (n = 52). For improved clarity, a logarithmic plot was chosen for the *y*-axis in (**A**,**B**). Increase (“delta”) in the neutralization capacity of VOCs BA.5, BQ.1.1, and XBB.1.5 after the 5th vaccination (**C**). Variant BA.5 was neutralized best both before and after the 5th vaccination. The increase in neutralization capacity was significantly greater for variant BA.5 compared to variants BQ.1.1 and XBB.1.5. The increase in neutralization of variant XBB.1.5 was significantly greater than that of variant BQ.1.1. Friedman test and Dunn’s multiple comparisons test as the post hoc test were performed for statistics. Adjusted *p* values are given.

**Figure 5 vaccines-12-00308-f005:**
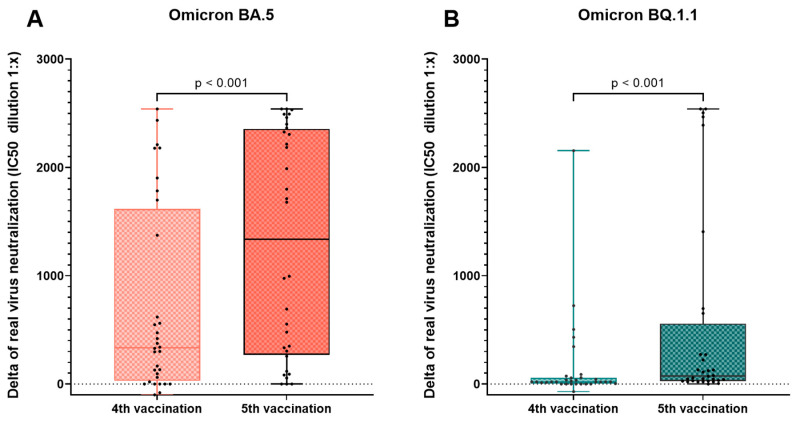
Change in the neutralization capacity of Omicron BA.5 and BQ.1.1 after vaccinations 4 and 5 in infection-naïve hemodialysis patients (n = 32). The delta (value after vaccination minus value before vaccination) in the neutralization capacity of each individual before and after the 4th (**A**) and the 5th (**B**) vaccination was calculated. A significantly bigger increase in neutralization capacity was seen for Omicron BA.5 and Omicron BQ.1.1 after the 5th compared to after the 4th vaccination. Statistical comparison was done using Wilcoxon test for connected samples. Blood analyses after vaccinations were done in the median 26 days after the 4th and 28 days after the 5th vaccination. Blood analyses before the respective vaccinations were done in a median of 124 days after the 3rd and 191 days after the 4th vaccination. We did not see a significant difference in baseline levels of neutralization capacities before the vaccinations for variant BQ.1.1 (*p* = 0.741); however, a trend of higher baseline values was seen before the 4th compared to before the 5th vaccination for variant BA.5 (*p* = 0.052). Median IC50 values before the 4th vaccination and before the 5th vaccination were 89.4 (20.0–588.7) and 67.35 (21.1–266.7) for BA.5 and 20.0 (0.0–20.0) and 20.0 (0.0–20.0) for BQ.1.1.

**Figure 6 vaccines-12-00308-f006:**
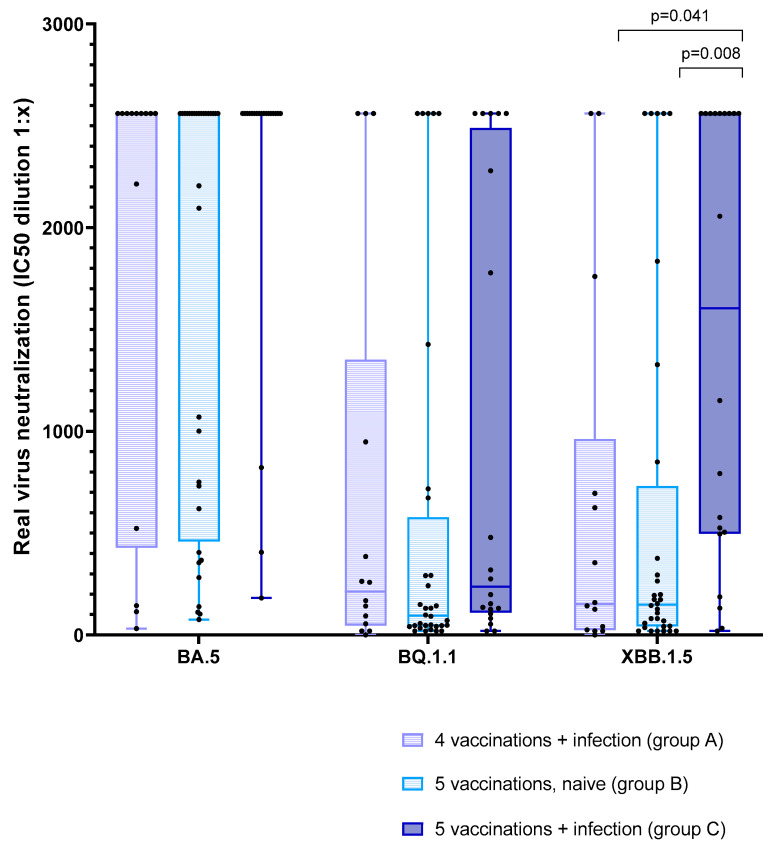
Real-virus neutralization capacity of Omicron variants BA.5, BQ.1.1, and XBB.1.5 in the whole cohort at the 4th blood analysis (n = 66). Comparison of live-virus neutralization capacities of the recently predominant Omicron variants in Germany BA.5, BQ.1.1, and XBB.1.5 in the three groups (A–C) of fourfold-vaccinated/infected (group A in pink) compared to fivefold-vaccinated/naïve (group B in blue) and to fivefold-vaccinated/infected (group C in purple) hemodialysis patients at the time of the 4th blood analysis (n = 66). For variant XBB.1.5, a significantly higher neutralization capacity was seen in those patients with 5 vaccinations and a breakthrough infection (group C), compared to groups A and B. Statistical comparison of the groups were performed using a Kruskal–Wallis test for each variant and a corrected Dunn’s multiple comparisons test as the post hoc test. Adjusted *p* values are given.

**Table 1 vaccines-12-00308-t001:** Patient characteristics.

	Total n = 66	4 Vaccinations + Infection (Group A) n = 14	5 Vaccinations, Infection-Naïve (Group B) n = 32	5 Vaccinations + Infection (Group C) n = 20	*p*
Age (years)	72.7 (60.3–81.5)	69.7 (58.5–79.5)	74.2 (64.8–82.1)	75.5 (59.8–80.6)	ns
Female	21 (31.8%)	2 (14.3%)	11 (34.4%)	8 (40%)	ns
Dialysis vintage (months)	42.0 (15.5–65.5)	17.2 (8.5–32.6)	55.4 (34.2–79.0)	33.6 (14.8–65.6)	*p* = 0.0023 (A vs. B) *
Immunosuppressive medication	10 (15.2%)	2 (14.3%)	5 (15.6%)	3 (15%)	na
Charlson Comorbidity Index [18]	7.0 (5.0–9.0)	7.5 (5.2–8.8)	8.0 (6.0–9.0)	6.0 (5.0–9.0)	ns
Renal diagnosis					na
Glomerulopathy	12 (18.2%)	3 (21.4%)	7 (21.9%)	2 (10.0%)	
Diabetic nephropathy	12 (18.2%)	4 (28.6%)	5 (15.6%)	3 (15.0%)	
Hypertensive nephropathy	8 (12.1%)	1 (7.1%)	5 (15.6%)	2 (10.0%)	
Congenital or cystic renal disease	7 (10.6%)	1 (7.1%)	4 (12.5%)	1 (5.0%)	
Tubulointerstitial disease	1 (1.5%)	0 (0.0%)	1 (3.1%)	0 (0.0%)	
Reflux nephropathy	0 (0.0%)	0 (0.0%)	0 (0.0%)	0 (0.0%)	
Other	6 (9.1%)	1 (7.1%)	2 (6.3%)	3 (15.0%)	
Nephropathy of unkown origin	20 (30.3%)	4 (28.6%)	8 (25.0%)	8 (40.0%)	
History of kidney transplantation	5 (7.6%)	2 (14.3%)	3 (9.4%)	0 (0.0%)	na

Results are presented as mean (±SD) and median (interquartile range) for normally and non-normally distributed data, respectively; categorical data are represented as total number (percentage). *p* values are given for χ^2^ test for categorial variables and adjusted *p* values are given for comparison with Kruskal–Wallis Test and Dunn’s multiple comparisons test as post hoc test corrected for multiple comparisons for continuous variables. Only significant *p* values are indicated. Abbreviations: ns, not significant; na, not applicable due to low numbers in subgroups. * post hoc group comparison for group A vs. B.

**Table 2 vaccines-12-00308-t002:** Comparison of immunity.

	4 Vaccinations + Infection (Group A)	5 Vaccinations, Infection-Naïve (Group B)	5 Vaccinations + Infection (Group C)	Group-Wise Comparison (*Kruskal–Wallis Test and Dunn’s test*)		
				*p* (*Adjusted p*) A vs. B	*p* (*Adjusted p*) A vs. C	*p* (*Adjusted p*) B vs. C
Neutralizing antibodies (BAU/mL)						
Before 5th vac./analysis 3	≥800 (147.5–≥800)	773.5 (506.2–≥800)	≥800 (675.2–≥800)	0.784	0.508	0.321
After 5th vac./analysis 4	≥800 (≥800–≥800)	≥800 (≥800–≥800)	≥800 (≥800–≥800)	0.088	0.925	0.052
anti-spike (S) IgG (BAU/mL)						
Before 5th vac./analysis 3	9205.0 (2387.5–34,979.5)	7948.5 (4310.2–23,892.5)	34,197.5 (10,497.2–≥40,000)	0.644	0.162	0.058
After 5th vac./analysis 4	36,955.5 (20,258.5–≥40,000)	≥40,000 (35,322.2–≥40,000)	≥40,000 (≥40,000–≥40,000)	0.121	**0.046**	0.388
Virus neutralization of Omicron BA.5 (IC50)						
Before 5th vac./analysis 3	130.3 (68.4–461.5)	67.3 (23.2–258.2)	286.9 (70.8–≥2560)	0.276	0.369	**0.037**
After 5th vac./analysis 4	≥2560 (945.6–≥2560)	≥2560 (566.3–≥2560)	≥2560 (≥2560–≥2560)	0.655	0.138	**0.023**
Virus neutralization of Omicron BQ.1.1 (IC50)						
Before 5th vac./analysis 3	20.0 (5.0–111.4)	20.0 (0.0–20.0)	47.2 (0.0–308.1)	0.385	0.380	**0.026**
After 5th vac./analysis 4	213.9 (64.6–807.7)	95.0 (45.0–387.6)	236.6 (120.7–2349.2)	0.458	0.571	0.072
Virus neutralization of Omicron XBB.1.5 (IC50)						
Before 5th vac./analysis 3	20.0 (0.0–44.5)	20.0 (0.0–20.0)	32.1 (15.0–175.7)	0.562	0.301	**0.010**
After 5th vac./analysis 4	151.2 (29.3–678.1)	149.5 (43.4–494.6)	1603.5 (502.3–≥2560)	0.948	**0.017** **(0.041)**	**0.002 (0.008)**

Results are presented as the median (interquartile range). *p* values are given for the group-wise comparison (Mann–Whitney test, exact *p* values, using GraphPad prism 9.5.1 (GraphPad Software, San Diego, CA, USA)) of patients with 4 vaccinations plus history of infections, patients with 5 vaccinations without infection, and patients with 5 vaccinations plus history of infection. Adjusted *p* values are given in brackets for comparison with the Kruskal–Wallis Test and Dunn’s multiple comparisons test as the post hoc test when significant. Abbreviations: vac, vaccination; min, minimum; max, maximum; BAU, binding antibody units; IC50, titer of 50% infection inhibition in a real-virus neutralization assay. Statistically significant values are bold-printed.

**Table 3 vaccines-12-00308-t003:** Immunity at 4th blood analysis of hemodialysis patients in the three subgroups stratified by the number of vaccinations and time of infection.

Group	Infection Subgroup	n	IgG (AU/mL)	NAb (BAU/mL)	Anti-Spike (S) IgG (BAU/mL)	BA.5 (IC50)	BQ.1.1 (IC50)	XBB.1.5 (IC50)
4 vaccinations + infection (group A)	early	3	4	≥800	29,413	523	55	41
	late	3	7	≥800	≥40,000	2214	142	143
	very late	8	20	≥800	36,956	2560	606	428
5 vaccinations, infection-naïve (group B)	all	32	1	≥800	≥40,000	2560	95	150
5 vaccinations + infection (group C)	early	5	4	≥800	≥40,000	2560	153	792
	late	8	11	≥800	≥40,000	2560	237	1604
	very late	7	65	≥800	≥40,000	2560	320	2560

Results are presented as the median for groups (“all”) and subgroups of patients with 4 vaccinations plus history of infections, patients with 5 vaccinations without infection, and patients with 5 vaccinations plus history of infection. Subgroups are stratified by the time of infection in early, late, and very late infection subgroups. Abbreviations: AU, arbitrary uni; NAb, neutralizing antibodies; BAU, binding antibody units; IC50, titer of 50% infection inhibition in a real-virus neutralization assay.

## Data Availability

The datasets for this manuscript are not publicly available because written informed consent did not include wording on data sharing (German data protection laws) as stated previously. Reasonable requests to access the datasets should be directed to the corresponding author.

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
