# Peer review of "SARS-CoV-2 Neutralization Capacity in Hemodialysis Patients with and without a Fifth Vaccination with the Updated Comirnaty Original/Omicron BA.4-5 Vaccine"

_vaccines, 2024, doi:10.3390/vaccines12030308_

Round 1
Reviewer 1 Report
Comments and Suggestions for Authors
The manuscript titled "SARS-CoV-2 neutralization capacity in hemodialysis patients with and without a 5th vaccination with the updated Comirnaty Original/Omicron BA.4-5 vaccine" provides valuable insights into the immune response elicited by an additional, fifth dose of the COVID-19 vaccine in hemodialysis patients. The authors found that a fifth vaccination significantly boosts neutralizing antibodies and enhances the neutralization of Omicron variants BA.5, BQ.1.1, and XBB.1.5. Overall, the manuscript is well-written; however, some areas could be enhanced to improve clarity. I recommend that the manuscript be accepted after these minor revisions are implemented.
· I suggest the authors provide a detailed description of the criteria used for selecting hemodialysis patients. Clarifying this will enhance understanding of the study population and its applicability to a wider range of hemodialysis patients.
· A more detailed discussion on the timing of vaccinations and blood sample collections is also recommended.
· While the findings on neutralization capacity post-5th vaccination are significant, a more comprehensive discussion on their clinical relevance is recommended. Specifically, it would be useful to explore how the neutralization capacity affects patient outcomes such as infection rates and severity.
· The authors briefly mention limitations, but a more thorough discussion would strengthen the study. This should include considerations of the study's observational design, the potential for selection bias, and the small sample size's impact on the generalizability of the results
Reviewer 2 Report
Comments and Suggestions for Authors
I was invited to revise the paper entitled "SARS-CoV-2 neutralization capacity in hemodialysis patients with and without a 5th vaccination with the updated Comirnaty Original/Omicron BA.4-5 vaccine". It was a cohort study aimed to evaluate the immune response against different variants of sars-cov-2 among hemodialyzed patients.
The topic is relevant both for clinicians and public health.
The paper was well written and methodology was deeply described.
Major observations:
- Introduction section was too poor. Authors should describe the vaccination schedule proposed in Germany against Sars-Cov-2 and deeply describe the importance of this research for public health;
- Sample size estimation was lacking;
- In Table 1 Authors should test differences among study groups;
-Authors should clarify if patients were infected more than one time;
- Among discussions, Authors should compare their results with similar study recently published.
Reviewer 3 Report
Comments and Suggestions for Authors
The present manuscript analysed the role of vaccination in Sars-COV2 disease among patients on hemodialysis. The authors showed that a 5th SARS-CoV-2 vaccination with the adapted vaccine improves both wild-type specific antibody titers and neutralizing capacity of the current Omicron variants BA.5, BQ.1.1, and XBB.1.5 in hemodialysis patients, suggesting that an additional booster vaccinations with adapted vaccines will likely improve immunity towards current and original SARS-CoV-2 variants. Overall, the manuscript is well designed and conducted, discussing on a topic with limited scientific evidence to date.
Minor comments
1) table 1 should be implemented by comparisons between groups
2) are there any information about dialytic membranes? For example, PMMA membrane have shown to improve immunological status of patients on hemodialysis
Comments on the Quality of English LanguageNo major issues detected
Round 2
Reviewer 2 Report
Comments and Suggestions for Authors
Authors properly addressed my previous comments. The paper can be accepted.